# The Impact of Intellectual Capital and Ownership Structure on Firm Performance

Znar Ahmed, Muhammad Rosni Amir Hussin * and Kashan Pirzada

Tunku Puteri Intan Safinaz School of Accountancy (TISSA-UUM), Universiti Utara Malaysia (UUM), Sintok 06010, Kedah, Malaysia
* Correspondence: mohdrosni@uum.edu.my

**Abstract:** Even though several studies have been done on intellectual capital, ownership structure, and firm performance, their status has remained uncertain in developing countries like Malaysia. Prior studies have generally focused on a single industry and overlooked the input of all Malaysian non-financial firms. This study investigates the impact of intellectual capital, its components, and ownership structure on firm performance. This study employs a balanced panel data examination for the data of 409 non-financial firms from 11 sectors listed on Bursa, Malaysia for five years (2016–2020). The modified value-added intellectual coefficient model was applied to examine the effect of IC efficiency on firm performance. The empirical findings revealed that IC efficiency, human capital efficiency, structural capital efficiency, capital employed efficiency, and relational capital efficiency are positively and significantly related to firm performance. However, physical and structural capital is the most substantial element of intellectual capital efficiency in augmenting profitability. In addition, government and foreign ownership positively affect firm performance. The research will help managers, policymakers, and investors understand how IC investments increase performance and make prudent investment choices in government and foreign ownership firms.

**Keywords:** intellectual capital; ownership structure; firm performance; Malaysia; GMM

## 1. Introduction

Tangible assets are vital in achieving firm efficiency, especially in an industrial economy. However, there is currently a transition toward a knowledge economy in which intangible assets play a crucial role in improving firm efficiency and growing knowledge to build a competitive advantage (Janošević et al. 2013). Intangible assets impact economic development and performance and enhance firms' competitive competence (Smriti and Das 2018). Intangible assets, particularly intellectual capital (IC), refer to the knowledge assets that can build and enhance a firm's value (Dzenopoljac et al. 2017). IC is the most essential and sensitive factor influencing business performance in today's global and knowledge economy. IC quickly becomes a fundamental capital component and a crucial instrument for creating new economic value (Nadeem et al. 2017; Smriti and Das 2018). Extent studies have indicated that IC serves as an intangible asset of the company (Kasoga 2020; Xu and Wang 2019).

For many years, the position of IC has controlled the generation of wealth in businesses (Vishnu and Gupta 2014; Soetanto and Liem 2019). Despite the extensive literature on IC, the results of the studies are inconsistent, with a lack of emphasis on Malaysian firms (Lee and Mohammed 2014). Additionally, Malaysian firms are unfamiliar with measuring intangible assets or IC compared to tangible and physical assets (Poh et al. 2018). Many companies and sectors in Malaysia have not yet incorporated the measurement of IC in their business (Poh et al. 2018). Malaysia is predicted to grow its number of skillful workers in the knowledge economy. This phenomenon will push IC to the center of the organization's long-term competitive advantage (Hashim et al. 2017).

The advantages of IC may be seen in the changes in countries from an industrial to a knowledge economy (Kasoga 2020). In doing so, Malaysia changed from a country focused on agriculture to the manufacturing industry. As a result, Malaysia began laying the foundation for a knowledge economy in the mid-1990s (Kweh et al. 2019). Consequently, Malaysia started promoting a knowledge economy in the early 2000s, recognizing the importance of fostering sustainable economic growth. The Knowledge Economic Master Plan was initiated and published in 2001 to achieve sustainable economic growth through investments in IC to keep pace with the emerging global economy (Kweh et al. 2019). This master plan provided strategic direction in human capital, technology, and R&D (Kweh et al. 2019). IC is crucial to the Malaysian government's plans and efforts to develop a knowledge economy (Kweh et al. 2019).

According to the resource-based view theory (RBV), IC significantly influences firm performance. The RBV is the most widely selected theory perspective of IC's strategic management fields (Newbert 2007). In addition, RBV stressed the importance of intangible resources that include analysing IC to create sustainable competitive advantages for a company. On the other hand, agency theory argues that ownership structure assists in mitigating the conflict between management and shareholders (Jenson and Meckling 1976). Furthermore, the agency theory suggests that the ownership structure of the firms determines the amount of information disclosed.

A company's success depends on its corporate governance framework, specifically for its shareholders and investors. It is necessary not just for the individual company but also for the stability of the financial system and the economy (Hooy et al. 2020). One of the key aspects of corporate governance studies is ownership consideration. It changes over time as a business issues new shares or existing shareholders trade heavily in the market (Kao et al. 2018). Hence, ownership is a common feature, and being highly concentrated was a major contributor that drove Malaysia into the Asian financial crisis (Mohd Ghazali 2020). In addition, like in other developing nations, family, government, and individual stockholders dominate the business environment in Malaysia (Mohd Ghazali 2020).

Researchers have examined the relationship between IC and ownership structure and corporate performance (Kasoga 2020; Smriti and Das 2018; Rashid 2020). The present research employed financial performance as a traditional indicator to analyze firm performance. Moreover, evaluating a company's financial performance assists decision-makers in determining the effectiveness of various levels of business strategies. Additionally, return on assets (ROA) and return on equity (ROE) are widely accepted among the researchers (Nadeem et al. 2017; Kweh et al. 2019; Yao et al. 2019; Soetanto and Liem 2019; Xu and Li 2019; Kao et al. 2018). Many researchers considered ROA and ROE to be the essential profitability indicators for determining a company's financial performance (Zhang et al. 2021).

This study focuses on Malaysian firms because IC's measurement as an intangible asset is yet to be identified and understood. Furthermore, the literature also indicates that the association between IC and performance has yet to be thoroughly explored, particularly in Malaysia's non-financial firms. Hence, this study investigates the relationship between IC efficiency, its components, ownership structure, and firm performance. Thus, the current study would expand the research on IC efficiency as the main factor of competitive advantages and understand how IC contributes to Malaysian non-financial firms' performance. Moreover, the current study provides significant insight into this potential relationship and contributes to the literature in numerous ways. Hence, the current study's contributions are as follows. First, despite several available studies, empirical evidence on the impact of IC on firm performance is scarce in Malaysia, and most studies have been conducted in developed countries. Hence, the current study addresses a gap in the literature by empirically examining the relationship between IC and firm performance in Malaysia. Second, previous studies from Malaysia usually investigate the case of the financial sector or some other sectors individually. Furthermore, this study covers 409 non-financial companies divided into 11 industries in Bursa, Malaysia. The non-financial listed firms were selected for the study because non-financial sectors are considered the

engine of economic growth, especially in emerging economies like Malaysia. Additionally, in the knowledge economy, the construction, energy, health care, technological industries, etc., can build job opportunities and provide new income sources for skilled workers and innovative products.

Third, the current study enhances the body of knowledge by performing the modified value-added intellectual coefficient (MVAIC) model by adding one additional IC component, relational capital efficiency (RCE), a more comprehensive model for measuring IC efficiency. Fourth, ordinary least squares (OLS) or fixed-effects (FE) models produce inconsistent estimators. Additionally, to deal with the possible endogeneity issue, this study used panel two-step GMM (Nadeem et al. 2017; Soetanto and Liem 2019). Hence, this study will provide non-financial firms in Malaysia with a better knowledge of the impact of ownership structure in determining firm performance. Fifth, this study presents an opportunity for companies to build competitive advantages using IC and its components to enhance firms' performance. As a result, the study seeks to address a gap in the literature by investigating which IC components improve profitability and value creation in Malaysian non-financial firms. Finally, since IC is a crucial component of the Malaysian government's Knowledge Economic Master Plan, which was initiated in 2001, this study's findings will assist the government in achieving sustainable economic growth through investments in IC to keep pace with the emerging global economy.

## 2. Literature Review and Hypotheses Development

### 2.1. Intellectual Capital Efficiency and Firm Performance

Investing in knowledge and IC has become essential for a firm's competitive advantage and performance improvement. Knowledge assets are a vital resource for achieving company success. IC is a commonly used concept for characterising knowledge assets, and it is widely known as the most crucial source of value-generating and competitive advantage (Kweh et al. 2019; Smriti and Das 2018). Fareed et al. (2016) found human capital to be the most significant component of intellectual capital. Although researchers have tried to define IC in various ways due to its complex nature, there is no specific definition or classification of IC. IC is defined as a resource ability and competence that drives organisational performance and corporate value creation (Nadeem et al. 2017; Smriti and Das 2018; Yao et al. 2019). Moreover, IC's components would assist in a better understanding of what IC is and allow firms to manage and report to their stakeholders (Bontis 1998). In the literature, IC is separated into many components. For instance, most scholars and studies classified IC as human, structural, and relational capital (Kweh et al. 2019; Mohammad and Bujang 2019b; Smriti and Das 2018). For instance, human capital (HC) is an essential and strategic resource for gaining competitive advantages and impacts firms (Bontis 1998; Kweh et al. 2019). Structural capital (SC) refers to a company's system, database, and process, and helps support its employee and firm performance (Kasoga 2020; Smriti and Das 2018). Relational capital (RC) is an organisation's powerful ability to improve engagement with community stakeholders and external parties such as clients, creditors, and suppliers (Soetanto and Liem 2019).

The effect of IC on business performance is a clear phenomenon. The value-added intellectual coefficient model (VAIC) is a performance indicator proportionate to a company's efficiency (Kasoga 2020). Many research pieces have proven a strong and positive association between IC and their sub-components' effect on corporate performance. For instance, a study in the technology sector of five ASEAN countries by Nimtrakoon (2015) revealed a positive influence of IC on ROA. Nadeem et al. (2017) documented a positive relationship between VAIC and performance in BRICS-listed firms. Similarly, taking 390 Korean manufacturing firms, Xu and Wang (2018) posited that IC positively impacts ROA. Additionally, a recent study by Mohammad and Bujang (2019b) discovered that the IC strongly links ROA in the Malaysian finance sector. Similarly, Kweh et al. (2019) reported that IC positively impacts the firm performance of the top 200 Malaysian firms. Likewise, Kasoga (2020) found that VAIC is positively and significantly related to ROA. Mohammad et al. (2018) found that the VAIC significantly positively correlated with the ROA of Malaysian firms.

Mohammad and Bujang (2019a) examined a comparative study between the three sectors, i.e., finance, construction, and plantation in Malaysia. They documented that IC has an adverse effect on performance in the plantation sector than the two other sectors. They reported a significant positive relationship between IC and performance in construction and finance.

The main theory used in developing the hypotheses to explain IC's importance on firm performance is the resource view theory (RBV). The RBV theory highlights the firm's reliance on intangible resources (Barney 1991). The vital idea of RBV theory is its attempt to assess the extent of companies' internal resources as factors within firms that drive competitive advantage, which will lead to superior performance amongst firms (Barney 1991). Based on the RBV theory, the firm's resources should possess four attributes; valuable, rareness, inimitable, and non-substitutability, shortly known as VRIN, to achieve competitive advantages (Barney 1991). In addition, this analysis anticipates that IC will positively affect Malaysian non-financial companies. Moreover, MVAIC, which incorporates four IC components, is used in the current study. Thus, the first hypothesis is proposed below:

**H1:** *Modified value-added intellectual coefficient positively influences the firm performance.*

The present study also examines the association between the IC components and performance. Earlier research has indicated that each element impacts performance differently from one to the next. Sardo and Serrasqueiro (2017) reported that human capital efficiency (HCE) positively correlates with firms' financial performance and market value in 14 Western European countries. Additionally, Smriti and Das (2018) found a positive impact of HCE on productivity among listed firms in India. In addition, prior studies have found that HCE strongly affects firm performance (Mohammad and Bujang 2019b; Nadeem et al. 2017; Nimtrakoon 2015). Furthermore, Li and Zhao (2018) reported a significant positive effect of structural capital efficiency (SCE) on sales growth in Chinese firms. Additionally, Nadeem et al. (2017) stated that SCE significantly positively affected firms' profitability and market value. Moreover, scholars found that SCE positively influenced performance (Dzenopoljac et al. 2017; Smriti and Das 2018; Soetanto and Liem 2019).

Nadeem et al. (2018) reported a positive relationship between capital employed efficiency (CEE) and firm profitability in Australian firms, which means that CEE plays a vital role in firm performance and competitive advantages. Asif et al. (2020) recently reported that CEE contributes to firm performance in the energy sector in Malaysia. According to the authors, physical capital remains Malaysian companies' primary source of financial performance. Furthermore, CEE denotes the effectiveness of financial capital in the VAIC model; many past studies reported an effect of CEE on performance (Mohammad and Bujang 2019b; Nadeem et al. 2017; Soetanto and Liem 2019). Additionally, Xu and Wang (2019) studied Chinese textile companies and documented that relational capital efficiency (RCE) significantly and positively impacts profitability. Additionally, Xu and Wang (2018) concluded that RCE is positively related to firms' profitability in Korean Manufacturing Industry. Similarly, the recent study by Mohammad and Bujang (2019b) revealed that the RCE does not affect performance in Malaysian financial firms. As a result, this study looks into the influence of the elements of MVAIC on company performance. As a result of the above arguments, the present study proposed the following hypotheses:

**H1a:** *Human capital efficiency positively influences the firm performance.*

**H1b:** *Structural capital efficiency positively influences the firm performance.*

**H1c:** *Capital employed efficiency positively influences the firm performance.*

**H1d:** *Relational capital efficiency positively influences the firm performance.*

### 2.2. Ownership Structure and Firm Performance

Many past and recent studies on the effect of ownership structure on firm performance are primarily based on agency theory. According to agency theory, the expenses

of settling disagreements between owners and managers are incurred when ownership and management are separated (Jenson and Meckling 1976). Corporations benefit from the concentration of ownership because high shareholdings allow for closer monitoring of managers (Jenson and Meckling 1976). According to agency theory, the greater overlap between ownership and management should eliminate conflicts of interest, resulting in improved firm performance (Fauzi and Musallam 2015). This relationship has been a significant source of concern in countries worldwide, including Malaysia. In this brief overview, the current study examines a set of ownership structures (government and foreign) known to impact Malaysian-listed firms. A government-linked corporation may be under even more pressure to make large profits to justify its existence. Because of the high level of public accountability in this form of organisation, it is reasonable to assume that government firms will try harder to achieve the nation's expectations (Mohd Ghazali 2020).

According to Jenson and Meckling (1976), government ownership is an important means of aligning the interests of owners and managers and curbing agency conflicts. As a result, government participation in the form of share ownership may influence the extent of agency conflict between management and outside shareholders (Jenson and Meckling 1976). In addition, the agency conflict between the manager and the stakeholder will be low in government ownership. In contrast, it has been shown that government ownership is often linked to serious agency issues and inadequate monitoring duties (Song et al. 2015). According to the logic of agency theory, government ownership lowers business performance because state owners pursue various goals, some of which contradict those of other firm stakeholders (Aguilera et al. 2021).

Numerous studies have been done to determine whether government ownership affects corporate performance. Tran et al. (2014) offered evidence that increasing government ownership in large corporations boosts their ROA. Additionally, several studies reported a positive direction in government ownership with performance (Fauzi and Musallam 2015; Mohd Ghazali 2020; Tran et al. 2014). In contrast, it has been suggested that the government's ownership of firms is a mechanism for the government to capture revenues made by firms for the benefit of politicians and bureaucrats rather than for commercial reasons (Phung and Hoang 2013). Furthermore, a negative of government ownership arises because the objective of government ownership is a political motive rather than generating profits for companies (Ting and Lean 2015). As a consequence, organizations' incentives to implement effective governance methods were reduced, resulting in poor company performance. Therefore, the next hypothesis is:

**H2:** *Government ownership influences the firm performance.*

According to Dahlquist and Robertsson (2001), foreign stockholders play the same role as institutional investors. Furthermore, international investors often have fewer links with insiders than local investors, allowing them to monitor the situation (Chen et al. 2009). Douma et al. (2006) established that foreign ownership positively influenced company performance. Recently, Rashid (2020) demonstrated that foreign owners significantly positively impact firms' performance. Many studies have demonstrated that the presence of foreign shareholders improves business performance (Douma et al. 2006; Tian and Estrin 2008; Mishra and Ratti 2011). Accordingly, the following hypothesis is proposed in this study:

**H3:** *Foreign ownership influences the firm performance.*

### 3. Research Framework

The study examines the relationship between IC efficiency, its components, and ownership structure toward firm performance. Thereby, intellectual capital was used as an independent variable with its sub-dimensions (human capital efficiency, structural capital efficiency, relational capital efficiency, and capital employed efficiency) as well as ownership structure (government and foreign ownership). Additionally, firm performance is used as a dependent variable underpinned by the resource-based view and agency theory. Therefore, Figure 1 below highlights the research framework.

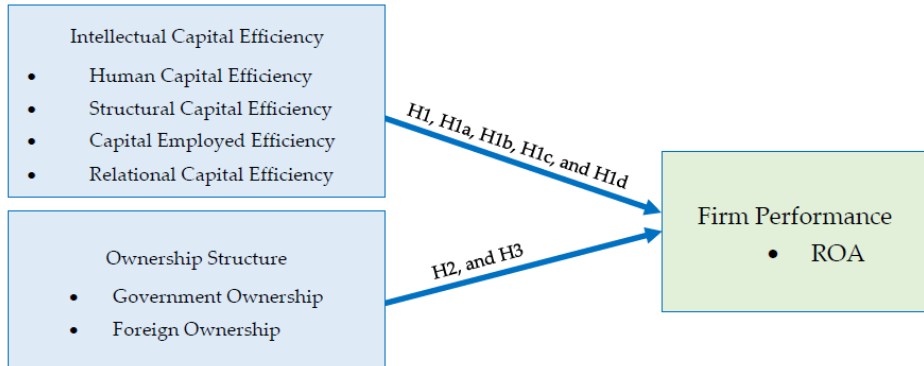

**Figure 1.** Theoretical framework.

## 4. Research Methodology

### 4.1. Sample and Data

The population of this study consisted of 785 firms listed on the Malaysian stock exchange (Bursa, Malaysia) from 2016–2020. Based on the purpose of this study, the current study excluded financial services firms' data. Furthermore, firms with missing data during the study period were also excluded to reduce bias and improve the reliability of the results. Additionally, firms with missing annual and combined annual reports were removed between two years. Moreover, the study's sample is balanced data of 409 firms (2045 observation) divided into 11 sectors, and the details are presented in Table 1. The present study obtained financial data from the Thomson Routers DataStream database. The data on government and foreign ownership variables were manually collected from annual public reports, which are available on the Bursa, Malaysia website.

**Table 1.** Sample classification.

| Sectors | Companies | Obs. | Percentage |
| --- | --- | --- | --- |
| Industrial products and services | 119 | 595 | 29% |
| Consumer products and services | 109 | 545 | 27% |
| Plantation | 32 | 160 | 8% |
| Technology | 28 | 140 | 7% |
| Construction | 27 | 135 | 7% |
| Energy | 25 | 125 | 6% |
| Transportation and logistics | 24 | 120 | 6% |
| Property | 17 | 85 | 4% |
| Utilities | 10 | 50 | 2% |
| Health care | 9 | 45 | 2% |
| Telecommunications and media | 9 | 45 | 2% |
| Total | 409 | 2045 | 100% |

### 4.2. Variable Measurement

#### 4.2.1. Performance Indicator

This study used the profitability indicator as the main firm's performance indicator. The reason for using return on assets (ROA) as a performance measure is that the ROA statistic informs investors about the efficiency with which the company transforms its investments into net income. Therefore, a greater ROA is preferable since the organization generates more money with less investment. The ROA measures a business's capacity to use its assets regardless of its financing approach. ROA is a standard accounting indicator for determining firm performance and is extensively used to assess a company's profitability (Nadeem et al. 2017; Mohammad and Bujang 2019b; Soetanto and Liem 2019; Xu and Li 2019). Furthermore, this study used ROE as a robustness check. The benefit of using ROE is that it explains the percentage of equity earned by the shareholders. Furthermore, investors

can use ROE to compare equity investment performance, hence helping them with their future investment decision strategy (Xu and Liu 2021).

$$ROA = Net\ Income/Total\ Assets \qquad (1)$$

### 4.2.2. Independent Variable

Several models have been suggested to measure IC due to the lack of consensus between scholars. Researchers have utilized the VAIC model to calculate the IC efficiency in academic and practical studies (Nadeem et al. 2017; Smriti and Das 2018; Soetanto and Liem 2019; Xu and Li 2019). The VAIC begins with calculating the value-added (VA):

$$VA = OUT - IN \qquad (2)$$

where (OUT) are all profits and revenues earned from producing goods and services, whereas (IN) are all costs (containing depreciation and amortisation) excluding employee's expenses, interests, dividends, and taxes (Mohammad and Bujang 2019b; Nimtrakoon 2015; Soetanto and Liem 2019). The above formula is called the total value-added approach. Moreover, the first component of IC is that HCE represents the powers of human capital (HC) to value creation in firms. Salaries, wages, benefits, and training expenses are a proxy for HC costs.

$$HCE = VA/HC \qquad (3)$$

SCE demonstrates how structural capital (SC) contributes to value development. Again, SC = VA − HC.

$$SCE = SC/VA \qquad (4)$$

The third component is CEE. A company's capital employed (CE) is derived by deducting its net assets from its intangible assets. CEE is an indicator of VA generated by a physical capital unit.

$$CEE = VA/CE \qquad (5)$$

The extra component is RCE for the original model of VAIC. However, relational capital (RC) is based on marketing costs such as selling and advertising (Soetanto and Liem 2019; Xu and Wang 2019).

$$RCE = RC/VA \qquad (6)$$

As discussed before, the formulation VAIC is calculated by the amount of the three components according to the following formula:

$$VAIC = HCE + SCE + CEE \qquad (7)$$

Finally, the MVAIC in the current study is the sum of all three original and new components.

$$MVAIC = HCE + SCE + CEE + RCE \qquad (8)$$

The current study measures ownership variables as follows. The percentage of government shares is utilized to calculate government ownership (Fauzi and Musallam 2015; Mohd Ghazali 2020; Tran et al. 2014). Additionally, the percentage of shares owned by foreign shareholders is used to calculate foreign ownership (Kao et al. 2018; Rashid 2020). The current study used two control variables: SIZE (Firm Size) is measured Ln of firms' total assets; AGE (Firm Age) implies the total years since its establishment (Kweh et al. 2019). The detailed list of variables is also given in Table 2.

**Table 2.** Variable Measurements.

| Variables/Abbreviations | Measurements |
|---|---|
| Dependent Variable | |
| Return on assets (ROA) | Net Income/Total Asset |
| Independent Variables | |
| Human Capital Efficiency (HCE) | VA/HC |
| Structural Capital Efficiency (SCE) | SC/VA |
| Capital Employed Efficiency (CEE) | VA/CE |
| Relational Capital Efficiency (RCE) | RC/VA |
| Modified VAIC (MVAIC) | HCE + SCE + CEE + RCE |
| Government Ownership (GVOWN) | The Proportion of Shares Owned by the Government |
| Foreign Ownership (FROWN) | The Proportion of Shares Owned by the Foreigners |
| Control Variables | |
| Firm Size (SIZE) | Ln of Total Assets |
| Firm Age (AGE) | Number of Years the Firm was Established |

### 4.3. Empirical Model

This section discusses the techniques for data analysis. Prior scholars Aslam and Haron (2020), Soetanto and Liem (2019), and Tran et al. (2020) claimed that the relationship between IC with firm performance bears the issue of endogeneity. In doing so, the results of OLS, fixed, or random-effects methods can be biased and incorrect (Baltagi 2005). Moreover, the current study performs the two-step system GMM. GMM was firstly utilized for dynamic panel data by Arellano and Bond (1991). The GMM approach supports a lag of the dependent variable and the lag of the independent variables as endogenous variables (Alghorbany et al. 2022; Nadeem et al. 2017; Yao et al. 2019; Sani et al. 2020). Moreover, the GMM estimator allows exogenous variables to be instrumented to manage with endogeneity and reduces the endogenous nature of data (Yao et al. 2019; Smriti and Das 2018). Furthermore, the system GMM method is more efficient in the presence of heteroscedasticity and serial correlation in the error terms (Arellano and Bover 1995). Additionally, the system GMM as proposed by Arellano and Bover (1995) and Blundell and Bond (1998) was more effective and appropriate than the GMM estimator, since it may employ level equations in addition to differenced equations to improve the efficiency of the results, particularly for data with a shorter time dimension. In addition, GMM produces biased results on panels with small T (time) and large N (observation) (Wintoki et al. 2012). Then, the following equations are given:

$$\text{ROA}_{it} = \alpha_0 + \beta_1 \text{ROA}_{it\text{-}1} + \beta_2 \text{MVAIC}_{it} + \beta_3 \text{GVOWN}_{it} + \beta_4 \text{FROWN}_{it} + \beta_5 X_{it} + \text{YEAR} + \text{INDUST} + \varepsilon_{it} \quad (9)$$

$$\text{ROA}_{it} = \alpha_0 + \beta_1 \text{ROA}_{it\text{-}1} + \beta_2 \text{HCE}_{it} + \beta_3 \text{SCE}_{it} + \beta_4 \text{CEE}_{it} + \beta_5 \text{RCE}_{it} + \beta_6 \text{GVOWN}_{it} + \beta_7 \text{FROWN}_{it} + \beta_8 X_{it} + \text{YEAR} + \text{INDUST} + \varepsilon_{it} \quad (10)$$

where $\text{ROA}_{it}$, is return on asset for the firm. $\text{ROA}_{it\text{-}1}$ is the one-year lagged firm performance. $\text{MVAIC}_{it}$ is a modified value-added IC. $\text{HCE}_{it}$ is human capital efficiency, $\text{SCE}_{it}$ is structural capital efficiency, $\text{CEE}_{it}$ is capital-employed efficiency, and $\text{RCE}_{it}$ is relational capital efficiency. $\text{GVOWN}_{it}$ is the government ownership, $\text{FROWN}_{it}$ is the foreign ownership, $X_{it}$ is the control variables, YEAR and INDUST are year and industry dummy variables, and $\varepsilon_{it}$ is a stochastic error term.

## 5. Results

### 5.1. Descriptive Statistics

Table 3 presents 2.95 percent as the mean value of ROA for the whole sample, which is smaller than the results reported by (Mohammad and Bujang 2019b). The average MVAIC score is 3.163, implying that for every monetary unit used, Malaysian non-financial enterprises produced an average of 3.163 from 2016–2020. Furthermore, the highest mean value is 1.898 for HCE, compared to RCE, SCE, and CEE, which are 0.602, 0.446, and 0.148, respectively. The average sum value of intangible components (HCE, SCE, and RCE) is 2.946; this is significantly greater than the average value of CEE of 0.148. This difference

implies that non-financial companies in Malaysia generate greater value from intangible assets than tangible ones. Regarding ownership structure variables, the mean of holding for foreign ownership (FROWN) and government ownership (GVOWN) are 4.77 and 8.958, respectively, with a minimum level of ownership of 0%, a maximum of 31.504% for the government ownership (GVOWN), and the foreign ownership (FROWN) was reported with minimum 0% and maximum 50.58%. Lastly, the mean value of SIZE (firm size) and AGE (firm age) is 13.232 and 36.594, respectively.

**Table 3.** Descriptive statistics.

| Variables | N | Mean | Min | Max | SD | Skewness | Kurtosis |
|---|---|---|---|---|---|---|---|
| ROA (%) | 2045 | 2.95 | −10.870 | 15.88 | 6.409 | −0.156 | 2.975 |
| MVAIC | 2045 | 3.163 | −2.002 | 7.733 | 2.073 | −0.284 | 4.252 |
| HCE | 2045 | 1.898 | −0.906 | 5.514 | 1.481 | 0.585 | 3.574 |
| SCE | 2045 | 0.446 | −0.843 | 1.596 | 0.517 | −0.351 | 4.185 |
| CEE | 2045 | 0.148 | −0.028 | 0.418 | 0.117 | 0.673 | 2.857 |
| RCE | 2045 | 0.602 | −1.638 | 2.776 | 0.895 | 0.023 | 4.675 |
| GVOWN | 2045 | 4.77 | 0.000 | 31.504 | 8.708 | 2.013 | 5.981 |
| FROWN | 2045 | 8.958 | 0.000 | 50.58 | 13.68 | 1.953 | 5.855 |
| SIZE | 2045 | 13.232 | 6.073 | 19.016 | 1.546 | 0.718 | 4.099 |
| AGE | 2045 | 36.594 | 1.000 | 137 | 19.643 | 1.758 | 7.822 |

### 5.2. Correlation

Table 4 shows the outcomes of the correlation test. The result shows that none of the coefficients is greater than 0.9, suggesting no severe correlation in the dataset (Hair et al. 2013; Pallant 2011; Tabachnik and Fidell 2012). Therefore, multicollinearity does not pose a threat to the estimation variables. In particular, the highest correlation is 0.85, which is between HCE and MVAIC. Table 4 also shows that MVAIC, HCE, SCE, CEE, GVOWN, FROWN, SIZE, and AGE positively correlate with ROA. As shown in Table 4, the variance inflation factor (VIF) value of all variables is between 1.09 and 6.33, and it is less than 10, so there is no multicollinearity issue (Hair et al. 2013; Tabachnik and Fidell 2012; Buallay 2018).

**Table 4.** Correlation matrix.

| Variables | 1 | 2 | 3 | 4 | 5 | 6 | 7 | 8 | 9 | 10 | VIF |
|---|---|---|---|---|---|---|---|---|---|---|---|
| 1. ROA | 1 | | | | | | | | | | |
| 2. MVAIC | 0.49 *** | 1 | | | | | | | | | 4.52 |
| 3. HCE | 0.54 *** | 0.85 *** | 1 | | | | | | | | 6.33 |
| 4. SCE | 0.09 *** | −0.09 *** | 0.20 *** | 1 | | | | | | | 3.16 |
| 5. CEE | 0.53 *** | 0.41 *** | 0.44 *** | −0.02 | 1 | | | | | | 1.49 |
| 6. RCE | 0.02 | 0.35 *** | 0.01 | −0.86 *** | 0.03 | 1 | | | | | 2.77 |
| 7. GVOWN | 0.08 *** | 0.22 *** | 0.28 *** | 0.11 *** | 0.10 *** | −0.06 *** | 1 | | | | 1.46 |
| 8. FROWN | 0.17 *** | 0.08 *** | 0.10 *** | 0.03 | 0.17 *** | −0.02 | −0.06 ** | 1 | | | 1.13 |
| 9. SIZE | 0.19 *** | 0.29 *** | 0.42 *** | 0.17 *** | 0.03 | −0.01 *** | 0.47 *** | 0.15 *** | 1 | | 1.80 |
| 10. AGE | 0.08 *** | 0.05 * | 0.01 | −0.07 ** | 0.01 | 0.09 *** | −0.01 | 0.25 *** | 0.12 *** | 1 | 1.09 |

***, **, and * denote significance at the 1, 5, and 10% levels.

### 5.3. Regression Analysis

The two-step system GMM outcomes of the effect of IC and ownership structure on firms' performance are illustrated in Table 5. Hence, in the first and second models, the signs of lagged ROA positively and significantly affect the current year ROA among Malaysian non-financial firms. The analyses reveal a positive correlation between MVAIC and ROA. The findings support H1 that IC is positive and significant for non-financial enterprises in Malaysia. The implication is that firms with a higher MVAIC tended to be more profitable. The results suggest that IC is an essential contributor to corporate success in Malaysia. The findings support the RBV theory perspective of the firms that IC and its components are strategic assets with a favorable influence on a firm's performance. Furthermore, it has been proven that IC should also be acknowledged as the firm's significant investment in driving

sustainable growth. The findings align with studies (Kasoga 2020; Mohammad and Bujang 2019b; Nadeem et al. 2017; Smriti and Das 2018; Soetanto and Liem 2019; Xu and Li 2019).

**Table 5.** Results of GMM regression.

| Variables | Model 1 | Model 2 |
|---|---|---|
| $ROA_{t-1}$ | 0.346 *** (0.046) | |
| $ROA_{t-1}$ | | 0.231 *** (0.042) |
| MVAIC | 1.47 *** (0.283) | |
| GVOWN | 0.464 *** (0.175) | 0.219 * (0.128) |
| FROWN | 0.424 *** (0.122) | 0.164 ** (0.082) |
| HCE | | 1.466 *** (0.296) |
| SCE | | 7.008 *** (1.431) |
| CEE | | 23.774 ** (11.879) |
| RCE | | 4.959 *** (1.084) |
| SIZE | −1.322 ** (0.883) | −0.066(1.024) |
| AGE | 0.088 (0.15) | 0.018 ** (0.134) |
| Constant | 3.192 (12.791) | −16.2 (13.379) |
| Industry Dummy | Included | Included |
| Year Dummy | Included | Included |
| Observations | 1636 | 1636 |
| No. of Groups | 409 | 409 |
| No. of Instruments | 59 | 59 |
| AR (1) (*p*-value) | 0.000 | 0.000 |
| AR (2) (*p*-value) | 0.86 | 0.41 |
| Hansen J. (*p*-value) | 0.80 | 0.68 |
| Prob > F | 0.000 | 0.000 |

***, **, and * represent significance levels of 1, 5, and 10 percent.

Furthermore, the study estimate model 2 uses the individual components of MVAIC and ownership structure (Government and Foreign ownership) as explanatory variables. Regarding the components, model 2 revealed that the main features (HCE, SCE, and CEE) positively and strongly affected ROA. The findings support H1a, H1b, H1c, and H1d. These results are similar to those studies (Mohammad and Bujang 2019b; Nadeem et al. 2017; Smriti and Das 2018). According to the findings, physical capital appears to be the major element, as indicated in the table above. This finding concurs with Nadeem et al. (2017), who claimed that the significance of CE as the primary contributor to the value generation of firms in developing nations cannot be neglected. Moreover, HCE and SCE are valuable sources of increased performance for businesses. The HC is a major factor in investing in the knowledge of employees and skills to improve a company's ability to innovate on processes and products.

Similarly, the present study found a positive influence of RCE as an additional component of VAIC with profitability. This result is similar to the findings (Xu and Li 2019; Xu and Wang 2019). Firms that invest in RC can build relationships with their suppliers, partners, and customers, and develop their relational networks, which appear to be essential to enhancing firm performance. This result is inconsistent with (Mohammad and Bujang 2019b; Soetanto and Liem 2019), who documented no impact of RCE on performance. Concerning ownership structure, government ownership (GVOWN) had a positive and significant impact on the ROA in both models; this result is compatible with H2. It argues that GVOWN leads to a better system of governance and company performance, and GVOWN managers are motivated to monitor the performance of the companies. This result is similar to (Fauzi and Musallam 2015; Tran et al. 2014). Furthermore, using models 1 and 2, the coefficient of foreign ownership (FROWN) is significantly positively associated with ROA at 1% and 5%, respectively. This finding supports H3, indicating that FROWN positively impacts company success. Moreover, this outcome aligns with prior results (Bentivogli and Mirenda 2017; Kao et al. 2018; Musallam 2015). Regarding the control variables in both models, SIZE negatively impacts ROA in model 1 and has no impact in

the other model. Furthermore, the results point out that firm age (AGE) was insignificant with firm performance in the first model and significant in the second model.

Combining all the regressors as exogenous factors and the lag of the independent variables as endogenous variables to test the consistency of the instruments, Table 4 shows the outcomes of two initial diagnostic tests, which are listed below the findings of the GMM estimator. The results of the Hensen J test for overidentification suggest that the instruments for both models are valid, with probability values of 0.80 and 0.68, respectively. Similarly, AR (2) test revealed that the *p*-value for the first and second models are 0.86 and 0.41. As a result, in both models, the probability value of AR (2) is greater than 0.1, indicating that there is no serial connection. In addition, as shown in Table 5, the number of instruments in the present investigation is less than the number of groups. It implies that the instruments employed were reliable and acceptable (Roodman 2009).

### 5.4. Additional Robustness Check

The results are robust to firm performance indicator (ROA) with two model IC measures, i.e., MVAIC and its components. Furthermore, the robustness test was employed to examine the credibility and consistency of the main regression findings. In this section, a new measurement of firm performance was formed to ensure that the findings are not sensitive to alternative measures. Return on equity (ROE) was used as an alternative performance measurement. Moreover, the ROE shows the efficiency in generating profit from each dollar of shareholders' investment. The results of the alternative estimation are reported in Table 6. Interestingly, the alternative methods reveal similar outcomes that are almost related to the outcomes of the main analysis using ROE. Specifically, the coefficient of MVAIC and its components are positive and significant in both the robust and the main analysis. It indicates that IC efficiency increases Malaysian firm performance when measured in terms of ROE. Additionally, the findings revealed that the investment in each component of IC generates higher efficiency and thus improves profitability. Additionally, government and foreign ownership in Malaysian non-financial firms were found to have a significantly positively impact on profitability when measured by ROE.

**Table 6.** Results of GMM regression (robustness test with ROE).

| Variables | Model 1 | Model 2 |
|---|---|---|
| $ROE_{t-1}$ | 0.35 *** (0.048) | |
| $ROE_{t-1}$ | | 0.234 *** (0.045) |
| MVAIC | 2.643 *** (0.452) | |
| GVOWN | 1.271 *** (0.38) | 0.691 ** (0.299) |
| FROWN | 0.766 *** (0.129) | 0.24 (0.192) |
| HCE | | 3.713 *** (0.681) |
| SCE | | 15.995 *** (3.066) |
| CEE | | 39.033 * (20.082) |
| RCE | | 9.038 *** (2.21) |
| SIZE | −4.657 *** (1.39) | −1.263 (2.066) |
| AGE | −0.09 (0.341) | −0.123 (0.275) |
| Constant | −15.14 (17.61) | −12.57 (14.32) |
| Industry Dummy | Included | Included |
| Year Dummy | Included | Included |
| Observations | 1636 | 1636 |
| No. of Groups | 409 | 409 |
| No. of Instruments | 58 | 58 |
| AR (1) (*p*-value) | 0.000 | 0.000 |
| AR (2) (*p*-value) | 0.17 | 0.68 |
| Hansen J. (*p*-value) | 0.73 | 0.46 |
| Prob > F | 0.000 | 0.000 |

***, **, and * represent significance levels of 1, 5, and 10 percent.

## 6. Discussion

This study looks at the influence of IC, its components, and ownership structure on business performance. The findings of this study show that firms with strong IC efficiency were more likely to be successful and more profitable. The findings imply that IC has a significant role in business performance in Malaysia. The result adds to the literature on IC by indicating that value creation in Malaysian listed companies is significantly influenced by IC efficiency. This outcome was consistent with prior research conducted in Malaysia. For instance, Lee and Mohammed (2014) and Mohammad and Bujang (2019a) reported a positive and strong association between IC efficiency and profitability indicator ROA as a measurement of firm performance. Additionally, the findings are in line with studies (Kasoga 2020; Kweh et al. 2019; Mohammad and Bujang 2019b; Smriti and Das 2018; Soetanto and Liem 2019; Xu and Wang 2019).

According to IC component results, the finding implies that HCE and SCE are valuable sources of increased business performance. For example, the HC is a major factor for investing in the knowledge of employees and skills to improve a company's ability to innovate on processes and products. Additionally, SCE resources, including systems, databases, and software, are essential to the firm's profitability. In contrast, CEE is a primary contributor to the value generation of enterprises in developing nations that cannot be ignored. Another component of IC efficiency that can enhance the firm's performance is RCE. The finding revealed that firms that invest in RCE could build relationships with their suppliers, partners, and customers and develop their relational networks, which appear essential to enhancing firm performance. The above findings supported prior studies (Mohammad and Bujang 2019b; Nadeem et al. 2017; Xu and Li 2019; Xu and Wang 2019). Regarding the ownership structure, the findings in model 1 and 2 show that GVOWN leads to a better system of governance and company performance, and GVOWN managers are encouraged to monitor the performance. This finding is aligned with (Fauzi and Musallam 2015; Tran et al. 2014). Lastly, models 1 and 2 revealed that FROWN has a favorable influence on business performance. Moreover, this result is consistent with prior results (Bentivogli and Mirenda 2017; Kao et al. 2018; Musallam 2015).

## 7. Conclusions, Implications, Limitations, and Recommendations for Further Studies

The relationship between IC, ownership structure, and business performance has received considerable attention from academics and researchers worldwide in the past two decades. In the current study, MVAIC was used with added RCE as an additional component as a proxy to measure IC, foreign, and government ownership as proxies of ownership structure. Because earlier research relied on OLS, FE, or RE, it ignored the relationship's dynamic nature. However, to fix the potential issue of endogeneity between IC and performance, this study used the dynamic panel regression two-step system GMM. This study demonstrates that IC efficiency is essential for enhancing the company's performance. The empirical findings add to a current study by showing that IC plays a significant role in developing value for Malaysian companies. This empirical analysis confirms that IC enhances a firm's profitability when used correctly and efficiently.

Further, regarding checking the impact of individual elements of MVAIC, the findings of model two revealed that all four components, namely (HCE, SCE, CEE, and RCE), had a strong association with firms' profitability. Additionally, government ownership (GVOWN) positively influenced ROA in both models. Therefore, it aligns with past studies (Fauzi and Musallam 2015; Tran et al. 2014). In addition, foreign ownership had positive and significant effects on ROA. Similarly, the same results were reported in prior studies (Bentivogli and Mirenda 2017; Kao et al. 2018; Musallam 2015). From a practical perspective, this study would assist managers in using several IC components to detect, capture, and measure the numerous IC types that must not be overlooked to improve firm performance.

Furthermore, the findings could be helpful for potential investors who want to predict a firm's future IC efficiency before making an investment decision. Additionally, policymakers and business managers may utilise the study results as a beginning point to

design more effective methods for using IC resources to obtain competitive advantages. Furthermore, this study provides evidence to policymakers that increasing GVOWN and FROWN in Malaysian non-financial companies can improve corporate performance.

This study, like many others, has some limitations. Firstly, this study applies the MVAIC model for assessing IC efficiency. Future studies should consider the original VAIC model to test IC's accuracy measure. Additionally, further studies may be undertaken using alternative methodologies for assessing IC efficiencies, such as the Balanced Scorecard, Skandia navigator, and National IC Index. Second, the MVAIC model is still questionable and criticized in the literature. Therefore, this study suggests that further research should add other components, such as innovation and process capital, to better understand and improve the MVAIC model. Third, the sample of this study is restricted to non-financial companies only. Future research might expand the study by incorporating financial firms and comparing financial and non-financial listed firms in Malaysia. In addition, future studies might be undertaken by collecting samples from other Asian nations. Fourth, this study focused only on two proxies of ownership structure with firm performance. Future research needs to be extended to investigate other dominant ownership variables, such as family, local ownership, etc., with different performance indicators such as asset turnover and market-based performance like Tobin's Q. ratio and market/book value. Lastly, this study is limited to investigating the direct relationship between IC and firm performance. In this regard, further studies can investigate the moderating or mediating role of other variables, such as corporate governance mechanisms between IC and firm performance.

**Author Contributions:** The research was done independently. All authors have equally contributed to all phases and parts of the manuscript. All authors have read and agreed to the published version of the manuscript.

**Funding:** This research received no external funding.

**Institutional Review Board Statement:** Not applicable.

**Informed Consent Statement:** Not applicable.

**Data Availability Statement:** Available upon request.

**Conflicts of Interest:** The author declares no conflict of interest.

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
