# Peer review of "The Impact of Intellectual Capital and Ownership Structure on Firm Performance"

_jrfm, doi:10.3390/jrfm15120553_

Round 1
Reviewer 1 Report
Please see the attached report. Thank you.

Author Response
Thank you for reviewing our paper, really appreciate the reviewers’ comments, which helped us to improve our manuscript.
Comment #1. The authors could visualize the research framework/ conceptual model.
Response: - We have made changes to the research framework. We tried to make it more informative.
Comment #2. The study uses dynamic GMM to address endogeneity problems. As such, it will be useful to further explain why the instrument(s) can be justified and reliably applied into the current setting (i.e., the relations between IC efficiency and firm performance).
Response: - We thank the reviewer for his constructive comments. We have made changes to the instruments and justified.
Comment #3. The endogeneity problem can be omitted variable bias. For example, you should benchmark your models with some seminal papers in the field or refer to more recent literature. For instance, the authors could discuss the effect of CEO duality (see e.g., Tran et al., 2020).
Response: - Suggestions have been addressed in the updated manuscript.
Comment #4. The economic significance of the results should be calculated and discussed in the paper.
Response: - We have added information about economic significance and highlighted in red color.

Reviewer 2 Report
Major issues:
1. The link between intellectual capital and ownership structure should be explained better. Why these two variables (put in connection) and not others? Why only one study (and not two), one for each of these two variables?
2. The title - what is the meaning of the comma between intellectual capital and ownership structure? It suggest that ownership structure is a part of intellectual capital. This is not the case. I suggest a reformulation of the title.
3. The contribution of the study has to be more underlined in the introduction of the paper.
4. The authors state in the introduction (p. 1): "Despite the importance of IC, it has been ignored as an essential contributor to a company's financial performance (Nadeem et al., 2017)." This statement is not sustainable reading the list of papers dealing with the subject, including some papers cited by the authors in the list of references and the description of literature presented in Section 2. In my opinion, this statement can mislead the readers and it has to be reformulated. The list below is only exemplificative:
M-Y Yusliza, Jing Yi Yong, M. Imran Tanveer, T. Ramayah, Juhari Noor Faezah, Zikri Muhammad ,A structural model of the impact of green intellectual capital on sustainable performance, Journal of Cleaner Production, Volume 249,2020,119334, https://doi.org/10.1016/j.jclepro.2019.119334.
Silvia Sumedrea, Intellectual Capital and Firm Performance: A Dynamic Relationship in Crisis Time, Procedia Economics and Finance, Volume 6, 2013, Pages 137-144, https://doi.org/10.1016/S2212-5671(13)00125-1.
Mohammad hosein Chizari, Rasool Zare Mehrjardi, Mohammad Mirmohammadi Sadrabadi, Fatemeh Kalantar Mehrjardi, The impact of Intellectual Capitals of Pharmaceutical Companies Listed in Tehran Stock Exchange on their Market Performance, Procedia Economics and Finance, Volume 36, 2016, Pages 291-300,
5. At least one paragraph should discuss why is the Malaysian case of interest for academic community.
6. Some studies, especially for emerging markets, link the government ownership with agency problems, and corruption. An alternative hypothesis to H2 should be discussed, at least in some phrases.
7. A correlation coefficient of 0.85 is almost equal to 1. A VIF of 6.33 is very big. At least add some more references to support the idea that these coefficients are small. Ideally, some robustness checks are required.
8. ROA is used as proxy for performance is using assets (even it is not perfect in accord with financial literature). ROE is used as proxy for performance of shareholders' invested funds. Their meaning is different. Section 4.4 should take these issues into account when the results are discussed.
9. Net earnings (with an important impact on both ROA and ROE) can be subject to earnings management, especially in connection with agency problems. This issue should be discussed more in the text.
Minor issues:
1. In conclusions, the use of H1, H2 (not defined) should be avoided. Some readers can be interested in reading only the conclusion and they will not understand these formulations.
2. Financial literature documents the differences between financial and non-financial companies. A comparison between these kinds of companies should be made cautiously (p. 15).
Author Response
Thank you for reviewing our paper, really appreciate the reviewers’ comments, which helped us to improve our manuscript.
Comment #1. The link between intellectual capital and ownership structure should be explained better. Why these two variables (put in connection) and not others? Why only one study (and not two), one for each of these two variables?
Response: - The link has been further explained in the literature review section according to the referee’s comments and suggestions.
Comment #2. The title - what is the meaning of the comma between intellectual capital and ownership structure? It suggests that ownership structure is a part of intellectual capital. This is not the case. I suggest a reformulation of the title.
Response: - Title of the manuscript is updated.
Comment #3. The contribution of the study has to be more underlined in the introduction of the paper.
Response: - The contribution of the study has been expanded and improved.
Comment #4. The authors state in the introduction (p. 1): "Despite the importance of IC, it has been ignored as an essential contributor to a company's financial performance (Nadeem et al., 2017)." This statement is not sustainable reading the list of papers dealing with the subject, including some papers cited by the authors in the list of references and the description of literature presented in Section 2. In my opinion, this statement can mislead the readers and it has to be reformulated.
Response: - The statement was removed from the manuscript.
Comment #5. At least one paragraph should discuss why is the Malaysian case of interest for academic community.
Response: - The introduction has been improved according to the referee’s comments and suggestions.
Comment #6. Some studies, especially for emerging markets, link the government ownership with agency problems, and corruption. An alternative hypothesis to H2 should be discussed, at least in some phrases.
Response: - We thank the referee for the valuable comments, and suggestions regarding the manuscript. The government ownership section has been expanded according to the referee’s comments and suggestions.
Comment #7. A correlation coefficient of 0.85 is almost equal to 1. A VIF of 6.33 is very big. At least add some more references to support the idea that these coefficients are small. Ideally, some robustness checks are required.
Response: - Justifications have been added according to the referee’s comments and suggestions.
Comment #8. ROA is used as proxy for performance is using assets (even it is not perfect in accord with financial literature). ROE is used as proxy for performance of shareholders' invested funds. Their meaning is different. Section 4.4 should take these issues into account when the results are discussed.
Response: - We thank the referee for the valuable comments, and suggestions regarding the manuscript. The section has been improved and justified according to the referee’s suggestions.
Comment #9. Net earnings (with an important impact on both ROA and ROE) can be subject to earnings management, especially in connection with agency problems. This issue should be discussed more in the text.
Response: -Further elaborated in the manuscript.
Comment #10. In conclusions, the use of H1, H2 (not defined) should be avoided. Some readers can be interested in reading only the conclusion and they will not understand these formulations.
Response: - Changes have been made to in the conclusion section.
Comment #11. Financial literature documents the differences between financial and non-financial companies. A comparison between these kinds of companies should be made cautiously (p. 15).
Response: - We thank the reviewer for the suggestion.

Reviewer 3 Report
Dear Authros,
the aim of this manuscript is to assess the impact of firm-level measures of intellectual capital efficiency and ownership structure on its performances, measured by return-on-asset and return-on-equity indicators.
The issue is interesting and the paper is well written. However, some shortcomings need to be necessarily addressed.
1. The paper lacks a theoretical background analyzing the reasons behind the relationship of interest. This should be breafly addressed in the introduction and in more detail in the literature review.
2. It is not clear to me how the GMM has been implemented in the empirical investigation. I suppose that the paper adopts a difference GMM, developed by Arellano and Bond (1991) instead of the more appropriate system GMM, developed by Arellano and Bover (1995) and Blundell and Bond (1998). This is more suitable to address both the unobserved firm-level heteorgeneity and the endogeneity of regressors, not accounted for in the current version of the paper.
3. A partially related issue is industry-level heteorgeneity and time fixed effects that are not included in the manuscript.
4. It should be clarified the reason why ROA and ROA are selected as performance measures of the firm, while excluding other relevant aspects. Moreover, which is the correlation between ROE and ROA? The introduction should specify which are the indicators adopted in the paper.
Minor issues
a. What do you mean by "knoledgeable worlers" (line 45)?
b. The ROA should be expressed in percentages.
c. Line 390 should be "This result is similar to the findings of [...]"
d. The page number is always 5.
References.
Arellano, M. and S. Bond. 1991. Some tests of specification for panel data: Monte Carlo evidence and an application to employment equations. The Review of Economic Studies 58: 277-97.
Arellano, M. and O. Bover. 1995. Another look at the instrumental variable estimation of error-components models. Journal of Econometrics 68: 29-51.
Blundell, R., and S. Bond. 1998. Initial conditions and moment restrictions in dynamic panel data models. Journal of Econometrics 87: 115-43.
Author Response
Thank you for reviewing our paper, really appreciate the reviewers’ comments, which helped us to improve our manuscript.
Comment #1. The paper lacks a theoretical background analyzing the reasons behind the relationship of interest. This should be breifly addressed in the introduction and in more detail in the literature review.
Response: - We have extended the introduction and added information about a theoretical background.
Comment #2. It is not clear to me how the GMM has been implemented in the empirical investigation. I suppose that the paper adopts a difference GMM, developed by Arellano and Bond (1991) instead of the more appropriate system GMM, developed by Arellano and Bover (1995) and Blundell and Bond (1998). This is more suitable to address both the unobserved firm-level heteorgeneity and the endogeneity of regressors, not accounted for in the current version of the paper.
Response: - We thank the reviewer for this valuable comment. We have improved the empirical model section and added information about the system GMM based on the reviewers’ suggestions and provided references.
Comment #3. A partially related issue is industry-level heteorgeneity and time fixed effects that are not included in the manuscript.
Response: - We have added industry and year as dummies, now each of the "year and industry" dummy included in our analysis.
Comment #4. It should be clarified the reason why ROA and ROA are selected as performance measures of the firm, while excluding other relevant aspects. Moreover, which is the correlation between ROE and ROA? The introduction should specify which are the indicators adopted in the paper.
Response: - We have clarified the reason behind selecting indicators in introduction section as well as variable measurements section.
All the minor issues have been addressed in the updated manuscript.

Round 2
Reviewer 2 Report
I have appreciated the authors' responses and the new version of the paper.
Author Response
Dear Reviewer,
Thank you very much for your suggestions. Indeed it has improved our manuscript.

Reviewer 3 Report
The authors should indicate the lag of endogenous variables adopted to perform the system-GMM
Author Response
Thank you very much for your suggestion. The lag of endogenous variable has been indicated.
